# Stochastic resonance mediates the state-dependent effect of periodic stimulation on cortical alpha oscillations

Jérémie Lefebvre[1]*, Axel Hutt[2], Flavio Frohlich[3]

[1]Krembil Research Institute, Toronto, Canada; [2]FE12 - Data Assimilation, Deutscher Wetterdienst, Offenbach am Main, Germany; [3]Department of Psychiatry, University of North Carolina at Chapel Hill, Chapel Hill, United States

**Abstract** Brain stimulation can be used to engage and modulate rhythmic activity in brain networks. However, the outcomes of brain stimulation are shaped by behavioral states and endogenous fluctuations in brain activity. To better understand how this intrinsic oscillatory activity controls the susceptibility of the brain to stimulation, we analyzed a computational model of the thalamo-cortical system in two distinct states (rest and task-engaged) to identify the mechanisms by which endogenous alpha oscillations (8Hz–12Hz) are modulated by periodic stimulation. Our analysis shows that the different responses to stimulation observed experimentally in these brain states can be explained by a passage through a bifurcation combined with stochastic resonance — a mechanism by which irregular fluctuations amplify the response of a nonlinear system to weak periodic signals. Indeed, our findings suggest that modulation of brain oscillations is best achieved in states of low endogenous rhythmic activity, and that irregular state-dependent fluctuations in thalamic inputs shape the susceptibility of cortical population to periodic stimulation.

DOI: https://doi.org/10.7554/eLife.32054.001

*For correspondence: jeremie.lefebvre@uhnresearch.ca

**Competing interests:** The authors declare that no competing interests exist.

## Introduction

Periodic brain stimulation, such as repetitive transcranial magnetic stimulation (rTMS) and transcranial alternating current stimulation (tACS), can be used to engage cortical rhythms (*Fröhlich, 2015*). Such findings have raised the fascinating prospect of manipulating rhythmic brain activity in a controlled manner, thereby engaging neural circuits at a functional level to manipulate cognition and to treat disorders of the central nervous system (*Cecere et al., 2015*; *Fröhlich, 2015*; *Romei et al., 2016*; *Ezzyat et al., 2017*). Yet, the brain is not a passive receiver. Invasive and non-invasive brain stimulation differ in their effect on neuronal dynamics as a function of the state of the targeted network, a robust observation across a variety of stimulation modalities (*Neuling et al., 2013*; *Ruhnau et al., 2016*; *Alagapan et al., 2016*). Indeed, previous studies have reported the seemingly counterintuitive fact that the susceptibility of neural tissue to exogenous control (during stimulation for instance) is increased during task-engaged states (*Neuling et al., 2013*; *Ruhnau et al., 2016*; *Alagapan et al., 2016*). Task-engaged states are generally characterized by higher neuronal firing rates and by population responses that are predominantly asynchronous, and by a suppression of low-frequency electrical brain activity compared to what is seen in the rest condition (*Churchland et al., 2010*; *Rosenbaum et al., 2017*). By contrast, the rest state is characterized by strong endogenous oscillations, such as alpha oscillations (8 Hz–12 Hz), which reflect internally driven brain processes (*Pfurtscheller et al., 1996*; *Klimesch, 2012*). To compensate for these state-dependent differences in population activity, it has been suggested that stimulation parameters should be calibrated in a closed loop (*Boyle and Frohlich, 2013*; *Lustenberger et al., 2016*) in order for stimulation to be optimally effective. But the lack of understanding about the cause of these state-

dependent changes and how they interfere with brain stimulation remains one of the main limitations for the development of new paradigms that are based on the selective control of brain rhythms.

We formulated the hypothesis that the dominance of asynchronous activity in the task state — in the form of irregular fluctuations in neural activity — may enhance the susceptibility of cortical neurons to periodic stimulation. Specifically, we asked whether stochastic resonance (SR) was involved. SR is a phenomenon by which the presence of uncorrelated random fluctuations, or so-called 'noise', amplifies the salience of a weak periodic signal driving a non-linear system (*McDonnell and Abbott, 2009*). In absence of noise, a subthreshold signal is too weak to generate an output that reflects the input. For intermediate levels of noise, however, the superposition of the signal and the random fluctuations results in super-threshold — and thus detectable — dynamics. Further increase in noise typically causes the saliency of the signal to decrease. Such a link between SR and state-dependent effects of brain stimulation have been reported experimentally across various modalities (*Schwarzkopf et al., 2011*; *Silvanto and Cattaneo, 2017*), in which irregular waveforms have further been shown to improve the effect of rhythmic stimuli (*Wuehr et al., 2017*). However, it remains unclear how the task-related decrease in alpha oscillations is involved in enhancing the susceptibility of cortical neurons to entrainment by exogenous electric fields.

Computational approaches and modeling are poised to answer many of these fundamental questions, but have received little attention, especially with respect to the clinical applications of brain stimulation. By identifying the network mechanisms involved, modeling can be used to catalyze the development of more efficient stimulation protocols and customized clinical interventions. Using simulations, we have previously used a cortical network model to better understand how periodic stimulation interacts with resting-state alpha (8 Hz–12 Hz) activity, and found that multiple mechanisms where involved (*Herrmann et al., 2016*). Echoing the theory of non-linear oscillators in physics, stimulations with increasing amplitudes and/or frequencies were found to shape networks responses through different mechanisms: *resonance*, in which intrinsic oscillations are enhanced by stimuli with frequencies close to the network natural frequency; and *entrainment*, where the systems' dynamics become phase locked to the stimulation and thus adopt the same frequency. Entrainment occurs only for specific sets of stimulation amplitudes and frequencies (regions in stimulation parameters space called Arnold tongues). But what happens in the presence of state-induced changes in baseline activity?

To address these questions and to provide predictions about optimal entrainment conditions, we harnessed computational and mathematical techniques to analyze state-dependent effects on a network model of the thalamo-cortical system in the presence of periodic stimulation. We found that increased thalamic drive suppressed endogenous oscillations throughout the thalamo-cortical loop, leading to an asynchronous state characterized by intense spiking activity that was weakly synchronized. We defined regimes of low (with respect to high) thalamic drive and resting (with respect to task-engaged) brain state to represent two externally imposed experimental conditions. In presence of periodic stimulation, the transition from the rest to the task-engaged state was accompanied by an increased susceptibility to entrainment, as demonstrated by amplified power at stimulation frequency and phase alignment between cortical responses and stimulation. Specifically, we found that the oscillatory response of the system switched from alpha oscillations to stimulus-induced activity as thalamic drive was increased, i.e. during a transition from the resting state to the task state. Taken together, these results show that the thalamo-cortical loop implements a gain control mechanism that regulates the robustness of alpha oscillatory activity and, by doing so, modulates cortical susceptibility to rhythmic entrainment by stimulation.

## Results

To understand how the effect of periodic stimulation depends on brain state (*Alagapan et al., 2016*), we examined the hypothesis that cortical susceptibility to entrainment is controlled by subcortical (i.e. thalamic) populations that are reciprocally connected with the cortical networks targeted by stimulation. Given the important role of the thalamus in controlling cortical state (*Poulet et al., 2012*) and resting-state alpha activity (*Hughes et al., 2004*; *Hughes and Crunelli, 2005*; *Lorincz et al., 2009*), we investigated how increased thalamic inputs regulated cortical excitability and the power of endogenous alpha oscillations. By brain state, we here refer to the state of the

thalamo-cortical network, which is determined by the afferent sensory drive to thalamus in the model and parallels two different experimental conditions (i.e. rest and task-engaged). We developed a model of the thalamo-cortical network composed of cortical, thalamic relay and reticular populations that exhibit synchronous activity within the alpha band (see Materials and methods). In this model, cortical populations of excitatory and inhibitory cells share both feedforward and feedback connections with the thalamus. Sensory inputs (e.g. visual) are sent through the lateral geniculate nucleus (LGN) to cortical areas for further processing (*Poulet et al., 2012*), impacting the pattern of activity of cortical neurons. The model structure is illustrated in *Figure 1*, along with the steady-state responses generated by the different groups of neurons present.

To quantify the impact of thalamic drive on cortical activity and alpha oscillations — first without stimulation — we calculated both cortical and sub-cortical firing rates, and the power of endogenous oscillations as the input to the thalamus was increased (*Figure 2*). We did this to define quantitatively different brain states, i.e. resting state vs task-engaged state. For low input to the thalamus, all populations across the network were maintained in a highly synchronous state, where recurrent interactions generate strong firing-rate oscillations with an envelope of about 8 Hz. As can be seen in *Figure 1B*, the activity of the different populations is synchronized, and the phase difference is due to the presence of propagation delays from the thalamus to and back from the cortex.

Increases in thalamic inputs have a destabilizing effect on endogenous oscillations. While increasing thalamic drive also increased firing rates throughout the system, a gradual suppression of synchronous alpha activity could also be observed: the power of endogenous alpha oscillations decreased substantially. This increased thalamic drive also changed the spiking patterns of cortical neurons: spiking activity became irregular and asynchronous, in line with studies showing a transition towards decorrelated cortical dynamics at task-onset (*Pfurtscheller et al., 1996*; *Poulet et al., 2012*; *Churchland et al., 2010*). This negative correlation between alpha power and spike rate is further in line with other experimental findings (*Haegens et al., 2011*; *van Kerkoerle et al., 2014*). Taken together, these results demonstrate that changes in thalamic state are sufficient to destabilize

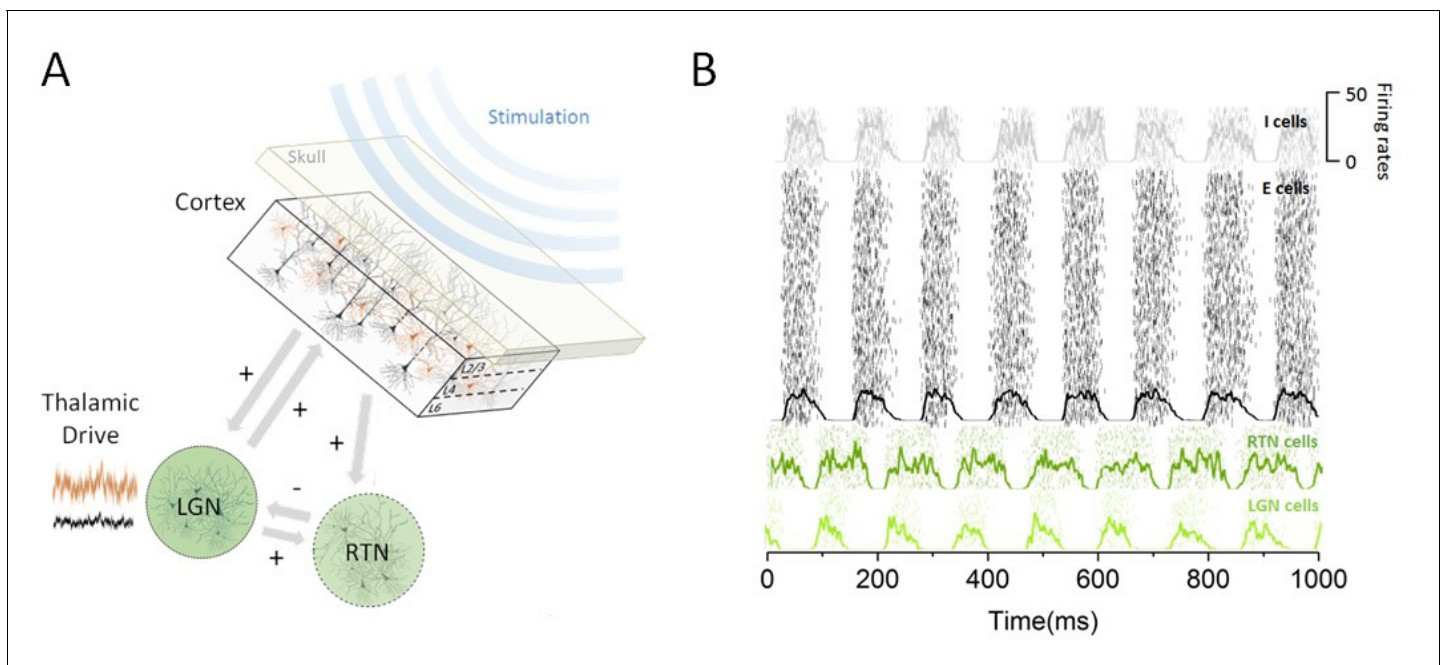

**Figure 1.** Thalamo-cortical circuit schematic and resting state activity. (**A**) Our computational model of the thalamo-cortical system is composed of both cortical (excitatory and inhibitory) and sub-cortical (lateral geniculate nucleus [LGN] and reticular [RTN]) populations. Sensory input, whose intensity scales with state, is assumed to arrive at the thalamus to then influence the dynamics of the whole circuit. Non-invasive periodic stimulation is applied only to cortical neurons. (**B**) In the resting state, both cortical and sub-cortical neurons are maintained in a deeply synchronous regime, displaying phase-locked firing rate oscillations in the alpha band (8 Hz). The spiking response of both cortical and thalamic populations is plotted for a single trial. Overlays represent the mean firing rate across neurons.
DOI: https://doi.org/10.7554/eLife.32054.002

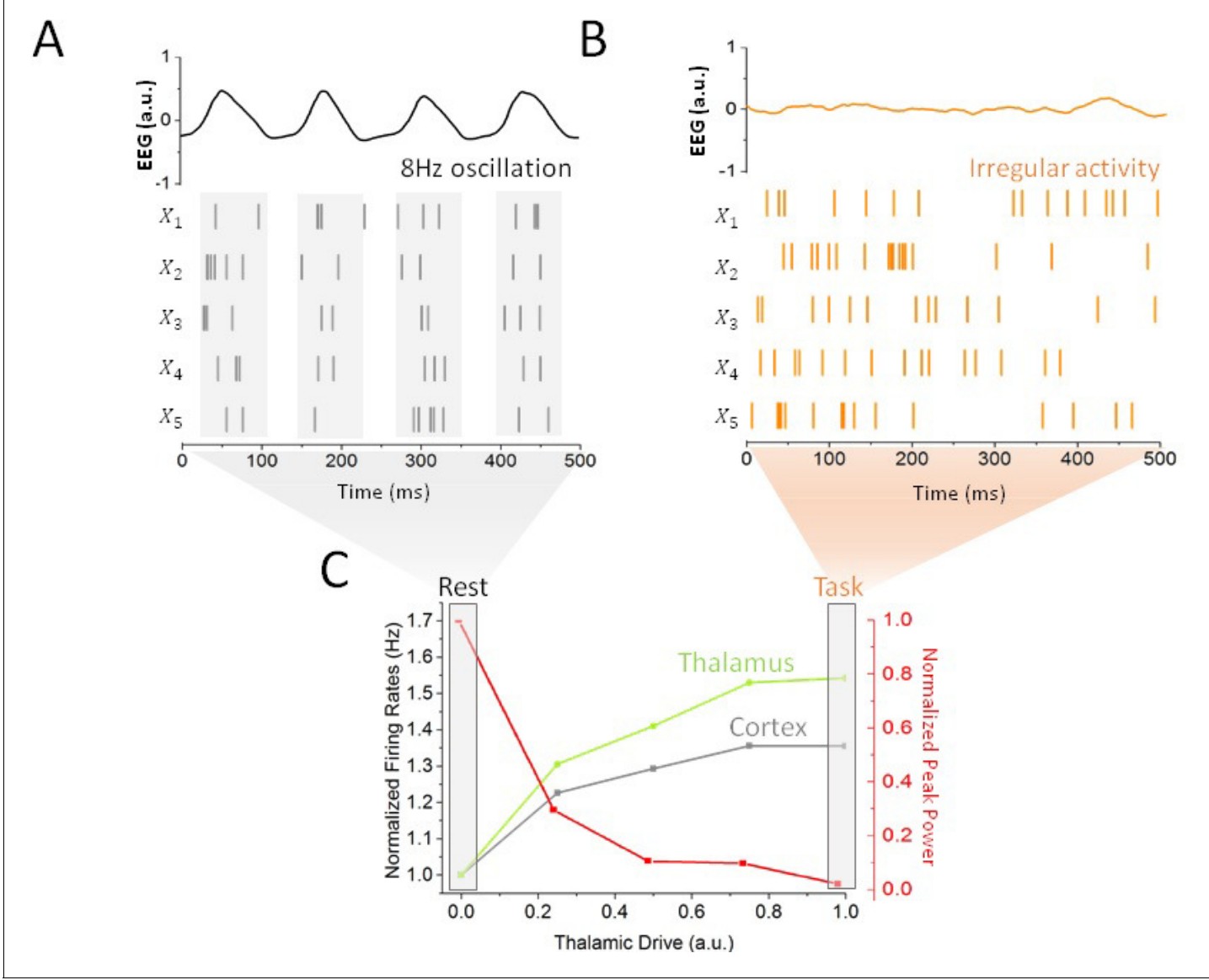

**Figure 2.** Impact of thalamic drive on firing rates and alpha power without stimulation. (A) Sample spiking activity of five randomly selected excitatory cortical cells in the resting state, which is defined as a regime of minimal thalamic drive ($D_{LGN} = 1x10^{-4}$). The firing activity of those neurons is highly correlated, and synchronized to the phase of simulated EEG alpha oscillations (top). (B) Activity of the same cortical neurons in the task state, where thalamic drive is high ($D_{LGN} = 1$). The spiking activity is now irregular and asynchronous, leading to a flat EEG activity where alpha oscillations have been suppressed. (C) Effect of gradual increase in thalamic drive on alpha power and mean firing rates for both cortical (grey) and sub-cortical (green) neurons.

DOI: https://doi.org/10.7554/eLife.32054.003

alpha oscillations throughout the thalamo-cortical loop, as well as to shape spike correlations. We thus defined rest and task states as limit cases of low (i.e. rest) and high (i.e. task-engaged) thalamic drive, respectively (see Materials and methods), as indicated in *Figure 2C*. We note that aside from the thalamic input, the network parameters were kept constant. Such a smooth and gradual destabilization of alpha oscillations is suggestive of a noise-induced supercritical Hopf bifurcation (See Materials and methods). According to this well-known mechanism (*Horsthemke and Lefever, 1982*), noise acts as a gain control parameter by which non-linear oscillations sustained by delayed and recurrent inhibition (as in our model) can be altered and/or suppressed by a change in noise variance (*Lefebvre et al., 2015*; *Hutt et al., 2016*). In the process, noise also linearizes the dynamics by

smoothing the non-linear response function of threshold systems (*Gammaitoni, 1995*; *McDonnell and Abbott, 2009*; *Lefebvre et al., 2015*). In the context of our model, rest and task-engaged states would then correspond to opposite points above and below the transition point, respectively, where the 'noise' is here mediated by increasingly irregular input from thalamus to cortex.

Having defined rest and task-engaged states in our network model and having characterized the influence of thalamic activity on alpha oscillations, we next asked how state-dependent changes in resting state oscillations (the destabilization of alpha oscillations) were impacted by cortical periodic stimulation. To disambiguate the contribution of resonance (the amplification of endogenous oscillations) and entrainment (the phase locking to an externally applied frequency) and to test network responses, we first considered a cortical stimulation with moderate amplitude and frequency of 11 Hz ($\omega_{stim}$), which does not share any low order harmonic relationship with the endogenous frequency (i.e. $\omega_o =8$ Hz). This was done to align our simulation to a previous experimental setting (*Alagapan et al., 2016*), in order to determine whether we could reproduce the previously observed state-dependent effects. Then, we investigated how spiking activity and cortical oscillations properties changed as thalamic drive was gradually increased in the presence of periodic stimulation (*Figure 3*). We first computed the correlations in the spiking response amongst excitatory cells in the cortical network, either with 11 Hz stimulation or with no stimulation (sham), see *Figure 3A*. Without stimulation, thalamic drive suppresses correlated spiking and synchronous activity decreases. By contrast, when stimulation is applied in regimes of weak thalamic drive, correlations are lower than in the sham condition, indicating that stimulation itself interferes with endogenous activity. By increasing thalamic drive, spike-rate correlations gradually increase, indicating that cells are becoming entrained by the stimulation. Taken together, these observations elucidate a gradual transition from recurrent to externally driven dynamics as thalamic input is increased. To explore this further, we plotted, in *Figure 3B*, the peak power found both at alpha frequency (8 Hz) and at stimulation frequency (here 11 Hz). The power at stimulation frequency did indeed increase during the transition from the rest to the task state and that a shift in oscillatory regime can be observed. Indeed, one can clearly observe a jump in the dominant frequency from 8 Hz (endogenous alpha frequency) to 11 Hz (stimulation frequency) as resting state alpha oscillations are gradually suppressed.

We hypothesized that this enhanced spectral power at the stimulation frequency might occur through SR. SR has been implicated in the enhancement of sub-threshold signals in which random fluctuations — of either external or internal origin — act like a pedestal increasing the sensitivity of stimulated neurons to a given set of low-intensity inputs (*Miniussi et al., 2013*). Initially formulated in bistable systems with an implicit time-scale (a 'resonance'), SR is also present in systems with threshold nonlinearities, which we note are present in our model.

To verify whether SR was indeed involved —and to disambiguate the contribution of SR and dithering (*McDonnell and Abbott, 2009*) — we chose a weak amplitude stimulation and computed the mutual information between stimulation signal and firing-rate response at the stimulation frequency across successive and independent trials, while noise variance was gradually increased across all populations in the network. As shown in *Figure 3D*, the mutual information was found to be optimized at an intermediate value, indicating that SR — in its most strict definition (*McDonnell and Abbott, 2009*) — was indeed involved. In contrast to SR, dithering — a similar phenomenon by which the saliency of weak signals is improved by noise — would cause the mutual information to increase monotonically, that is, it would not exhibit a peak at some intermediate value of noise. To confirm the presence of entrainment (to the stimulation and not to another internal oscillation), we performed a phase analysis comparing the phase of the cortical firing rates with the phase of the stimulation across independent trials. The results are shown in *Figure 3E and F*. There, one can see that weak phase alignment was found in the rest state (*Figure 3E*), whereas clear phase locking can be seen in the task state (*Figure 3F*). Our simulations show that cortical spiking responses are phase locked to the stimulation waveform only for intermediate values of thalamic drive. Below this drive, endogenous network activity suppresses cortical entrainment, and beyond this value, irregular noisy fluctuations originating from the thalamic population decreases the saliency of the temporal structure of the stimulation waveform. Taken together, the results above indicate that state-dependent inputs simultaneously amplify stimulation-induced activity while suppressing endogenous rhythmic activity through SR, enabling the stimulation to entrain cortical neurons.

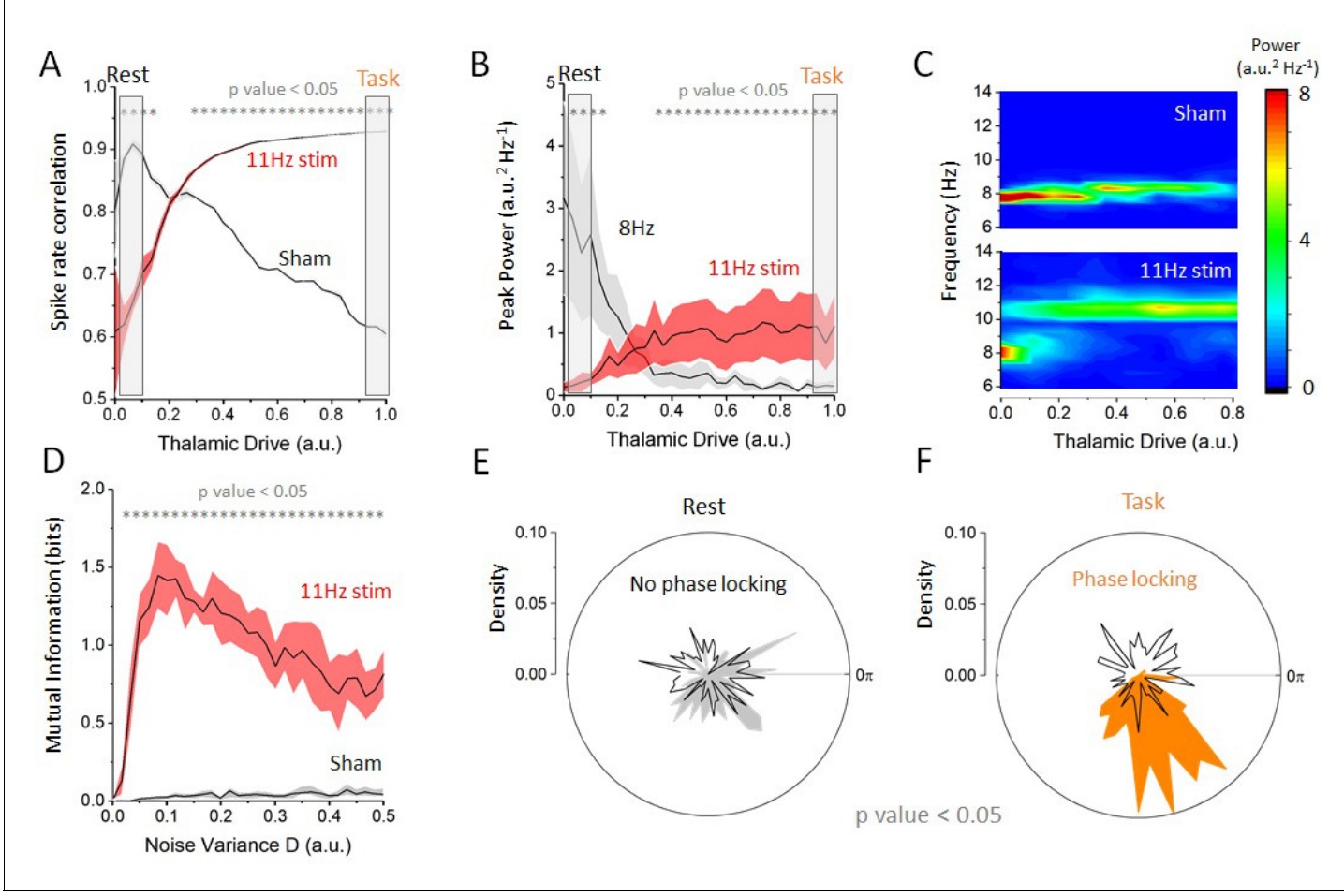

**Figure 3.** Effect of increasing thalamic drive on network dynamics with stimulation. (A) Firing-rate correlations of cortical excitatory neurons (average pairwise correlation between firing rates) as a function of thalamic drive with 11 Hz stimulation and sham (no stimulation). Without stimulation (sham), correlations in the network decrease as endogenous oscillations are suppressed. By contrast, with 11 Hz stimulation, correlations increase monotonically with thalamic input as the cells become entrained by the stimulation. (B) Peak power at the endogenous alpha frequency ($\omega_o$=8 Hz) and at the stimulation frequency ($\omega_{stim}$=11 Hz) as a function of increasing thalamic drive. The power of the endogenous oscillations is gradually suppressed, whereas the opposite occurs at the stimulation frequency, suggesting a transition between oscillatory regimes. (C) Full-power spectral density distribution as a function of thalamic drive. Without stimulation, increase in thalamic drive destabilizes endogenous alpha oscillations and power is gradually suppressed. In presence of stimulation, the destabilization of endogenous oscillations is more abrupt and is replaced by spectral power at the stimulation frequency. Stimulation parameters for (A), (B) and (C) were $S$ =0.15, $\omega_{stim}$=11 Hz. (D) Mutual information between stimulation and network response as a function of increasing noise throughout the network, both with and without stimulation. When stimulation of weak amplitude (here $S$ =0.10) is applied, and the mutual information peaks at some intermediate value of noise, indicating that stochastic resonance is involved. (E) Phase distribution of firing-rate responses across trials at 11 Hz in the rest state ($D_{LGN} = 1x10^{-5}$). In each trial, the stimulation was applied at a random phase. This is why the phase distribution is uniform over all angles (black line). The phase difference between the stimulation and the network response is also uniform, indicating that the network dynamics are not phase locked to the stimulus (i.e. there is no entrainment). (F) By contrast, in the task state ($D_{LGN} = 1x10^{-5}$), cortical firing rates are phase locked to the stimulation, and the distribution shows a strong peak at the preferred phase (orange area). Here, $S = 0.1$ and $\omega_{stim} = 11$ Hz.

DOI: https://doi.org/10.7554/eLife.32054.004

*Figure 4* summarizes these findings in the rest and task-engaged states. In the rest state (*Figure 4A*), the dynamics of the cortical neurons were predominantly governed by internal, recurrent fluctuations. When stimulation is applied in this state (*Figure 4B*), ongoing alpha oscillations are slightly perturbed but no cortical entrainment occurs. Stimulation has a minimal impact on the activity of cortical neurons: the phasic alternation of network-wide coincident firing and inhibitory recovery greatly reduces susceptibility to entrainment. This absence of phase locking of neuronal activity to the stimulation signal can be most clearly seen from the firing activity of cortical excitatory and

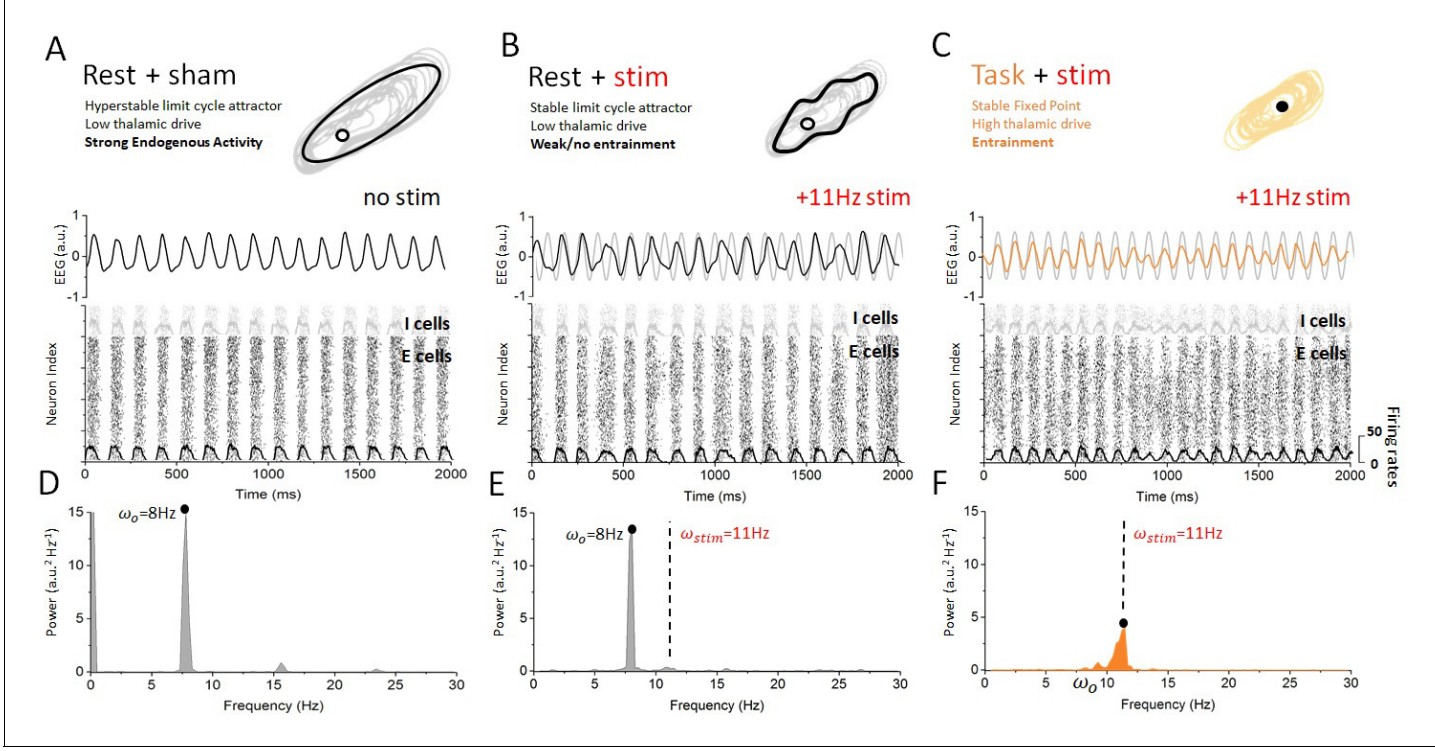

**Figure 4.** Dynamics of cortical neurons in the rest and task states with and without stimulation. (A) In the absence of stimulation, the resting state is characterized with strong synchronous alpha oscillations, leading to stable attractor dynamics. These oscillations correspond to a stable limit cycle surrounding an unstable fixed point. Both excitatory and inhibitory neurons display firing-rate modulations (black and gray overlays) at the endogenous frequency. (B) When stimulation is applied in the rest state, the interaction between correlated neural activity and stimulation-induced fluctuations in membrane voltage results in weak entrainment. Despite the presence of periodic stimulation impacting all cortical neurons equally, the dynamics of the simulated EEG are predominantly characterized by endogenous oscillations (top). The spiking activity of both excitatory and inhibitory neurons remains locked to endogenous cycles, where stimulation has little to no impact on network activity (bottom). (C) By contrast, endogenous oscillations are suppressed in the task state, where the dynamics now evolve around a fixed point. Simulated EEG activity is fully entrained to the stimulation (top), and so are cortical neurons whose spiking is phased locked to the stimulation frequency. (C) The power spectral density distribution of the EEG response in the rest state. The spectrum is largely dominated by endogenous oscillations ($\omega_o$ =8 Hz) and only weak contribution can be observed at the stimulation frequency ($\omega_{stim}$ =11 Hz), indicating that network oscillations are not entrained by the stimulation. (D) The power spectral density distribution of the EEG response in the task state. Here, by contrast, the power at the endogenous frequency has been almost fully suppressed and a clear peak can be seen at the stimulation frequency. This implies that network oscillatory activity is fully determined by the stimulation. Stimulation parameters here are $S = 0.15$, $\omega_{stim} = 11$ Hz.

DOI: https://doi.org/10.7554/eLife.32054.005

inhibitory neurons. The firing activity of these cells occurs by successive bursts at 8 Hz, with little effect from stimulation-induced membrane fluctuations, which remain sub-threshold. The poor susceptibility is also apparent from the relatively small contribution of stimulation power to the overall spectral density distribution of the simulated cortical EEG (*Figure 3C*). These indicate that the rest state is characterized by highly stable attractor dynamics. By contrast, robust entrainment was observed in the task-engaged state (*Figure 4C*), as seen from a dominant spectral peak located precisely at the stimulation frequency (*Figure 3C*). Despite the fact that stimulation amplitude and frequency remained the same, cortical neurons were here fully entrained by the 11 Hz stimulation. The irregular activity that characterizes the task state was here replaced by stimulus-induced fluctuations while neural firing was tightly phased locked to the stimulus dynamics. Spectral power at the stimulation frequency is significantly amplified when compared to that in the rest state, whereas negligible power can be found at the endogenous alpha frequency. This forms yet another indication that the resting state attractor has been destabilized under the action of thalamic drive and the resulting increase in neural noise (*Lefebvre et al., 2015*; *Hutt et al., 2016*), also in line with requirements of SR (*McDonnell and Abbott, 2009*).

But how does this state-dependent susceptibility depend on stimulation parameters? To answer this question, we first fixed stimulation amplitude and varied its frequency between 0 and 50 Hz, while computing the power spectral density in each case (*Figure 5*). A peak at the stimulation frequency can be seen along the diagonal, but the rest state power spectral density is characterized by a dominant horizontal peak at 8 Hz. According to this analysis, the spectral contribution of stimulation is negligible unless the stimulation frequency is close to the endogenous frequency and/or its harmonics (horizontal lines in *Figure 5A*), as expected via resonance (*Ali et al., 2013*; *Herrmann et al., 2016*). In the task-engaged state, the opposite occurs: reduced power at the endogenous frequency promotes entrainment, and stimulation power increases significantly across the range of stimulation frequency considered.

As a next step, we computed the power-spectral density of the simulated EEG activity of cortical excitatory and inhibitory cells, and systematically measured the peak frequency and power for all combinations of stimulation frequency and amplitudes within a given range. We then identified regions for which the dominant (i.e. peak) frequency was defined either by endogenous oscillations or by the stimulation — thus identifying regimes of entrainment. In *Figure 6A*, one can see the peak frequency of the simulated cortical EEG activity in the rest state. For most combinations of stimulation amplitudes and frequencies, the peak frequency remains stable and equal to the endogenous alpha frequency (i.e. 8 Hz). However, for higher amplitudes and for stimulation frequencies near 8 Hz, cortical neurons were gradually entrained by the stimulation (1:1 entrainment). A narrow triangular entrainment region (the so-called Arnold tongue, delimited by white dashed lines) (*Glass and Belair, 1980*; *Hunter and Milton, 2003*) emerges and gets larger as stimulation amplitude increases. Note that the asymmetrical shape of the Arnold tongue is here due to a stimulation-induced shift of the endogenous frequency (*Hutt et al., 2016*). The equivalent calculations were made in *Figure 6B* but for the task-engaged state. Multiple differences compared to the rest state can be observed. The Arnold tongue spans a much larger portion of the stimulus-parameter space, indicating robust entrainment for much weaker stimulation amplitudes. In *Figure 6C and D*, the power at peak response frequency is plotted. The power found at the endogenous frequency in the rest state outside the Arnold tongue (here also delimited by a white dashed line) is high, and is gradually

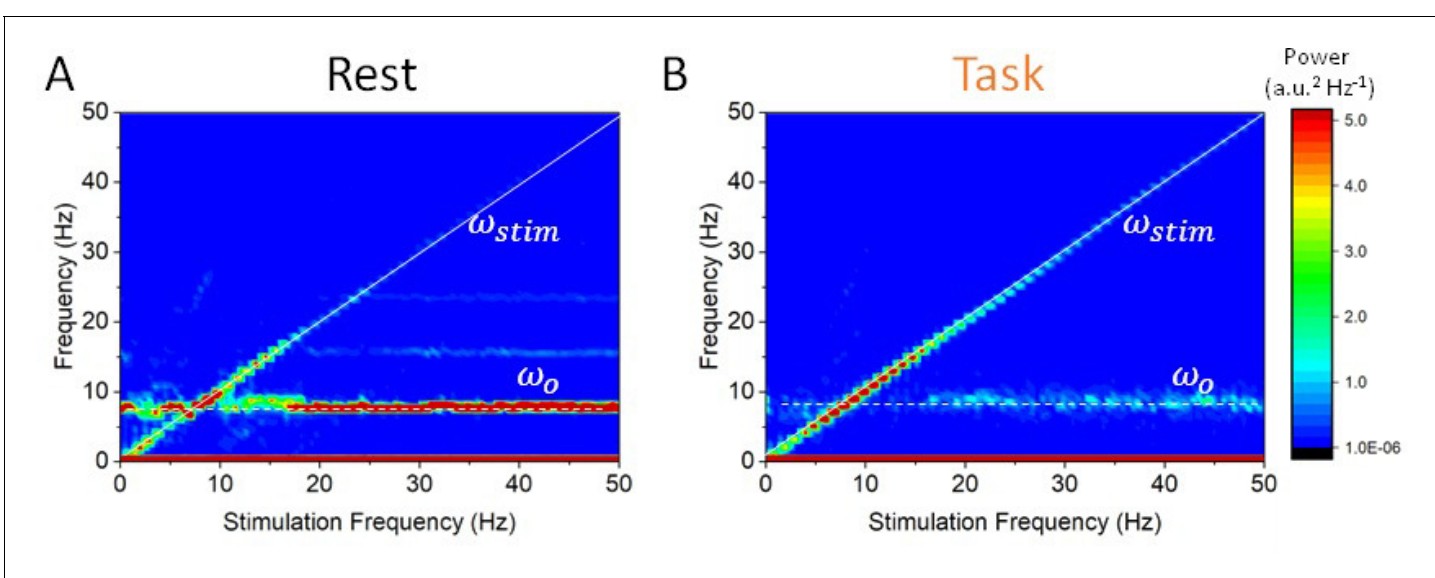

**Figure 5.** State-dependent responses to periodic stimulation with variable frequency. (**A**) Power in the rest state is concentrated at the endogenous frequency (horizontal dashed line at 8 Hz) while the stimulation frequency $\omega_{stim}$ is increased from 0 to 50 Hz. The peak power found at the stimulation frequency (dashed line along the diagonal) is small unless $\omega_o \approx \omega_{stim}$, that is, when the stimulation frequency is close to the peak alpha frequency. (**B**) In the task state, power is instead concentrated at the stimulation frequency across the range of values visited (along the diagonal). By contrast, power at the endogenous oscillation is much smaller. Note that the power at the stimulation frequency scales with the distance with respect to the alpha peak at 8 Hz. Stimulation parameters here are $S = 0.15$ while $\omega_{stim}$ was varied from 0 to 50 Hz.
DOI: https://doi.org/10.7554/eLife.32054.006

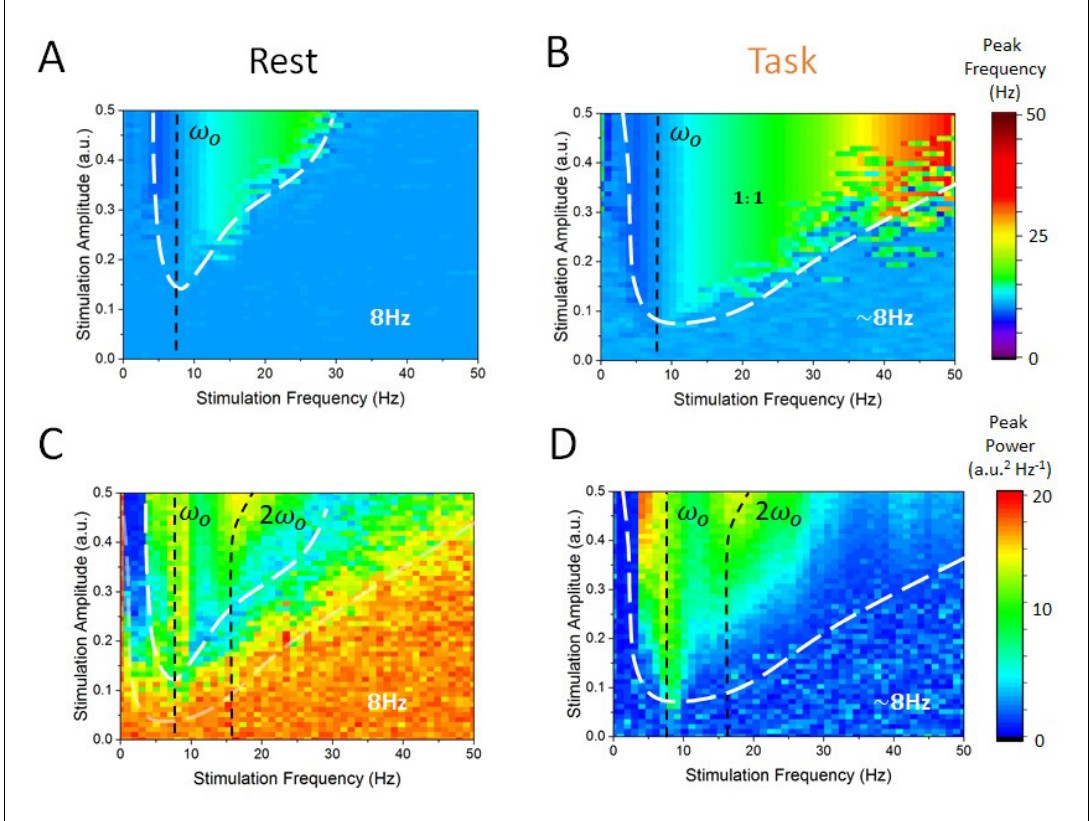

**Figure 6.** State-dependent Arnold tongues as a function of stimulation amplitude and frequency. By systematically varying the stimulation amplitude ($S$) and frequency ($\omega_{stim}$) and identifying the associated peak frequency (i.e. frequency where maximal power can be found) and peak power (i..e power at the peak frequency) in each case, we can delimit regions in parameter space where cortical dynamics are either governed by endogenous alpha oscillations or locked to the stimulation. (**A**) In the rest state, the vast majority of stimulation parameter space is characterized by an absence of entrainment. The peak frequency of cortical EEG activity remains at 8 Hz. However, as stimulation amplitude is increased for stimulation frequency near 8 Hz, entrainment occurs. However, this triangular region — called the Arnold tongue — remains narrow. (**B**) In the task state, the Arnold tongue increases significantly, and occupies most of the range of stimulation parameters considered — entrainment is thus more prevalent in the task state. (**C**) Peak power in the rest state for varying stimulation parameters. (**D**) Same as in (C) for the task state. Stimulation amplitude was varied within [0, 0.5] and its frequency within [0, 50 Hz].

DOI: https://doi.org/10.7554/eLife.32054.007

suppressed as stimulation parameters are changed towards the entrainment region. There, peak power, associated with the stimulation frequency is much smaller (*Figure 6C*). Under the action of thalamic input in the task state, endogenous alpha oscillations are suppressed compared to those in the rest state, leading to weak power outside the Arnold tongue (*Figure 6D*). In contrast to the rest state, stimulation at the endogenous frequency enhances the peak power. In sum, the effect of stimulation on peak power when the frequency is close to the resonant frequency (vertical black dashed lines in *Figure 6C and D*) also depends on the state, that is, the thalamic drive.

What mechanism is responsible for this state-dependence? Mathematical insights can play a key role here in helping us to understand how periodic stimulation interacts with intrinsic limit cycles (i.e. alpha oscillations) and how changes in stability can promote entrainment. In *Alagapan et al., 2016*), the authors modelled state-dependent entrainment using a population-scale network of cortical neurons interacting through both local and feedback projections in the presence of robust resting state oscillations. To learn more about the mechanism involved, we considered a simplified neural oscillator with delayed feedback (*Lefebvre and Hutt, 2013*; *Lefebvre et al., 2015*; *Hutt et al., 2016*) as a simplified model for the thalamo-cortical network. Networks of neurons that are exposed to delayed feedback and recurrent inhibition commonly display rhythmic activity whose features are tightly linked to input statistics (*Dhamala et al. 2001*; *Doiron et al., 2003*; *Lefebvre and Hutt,*

*2013*). To better understand the mechanism underlying the state-dependent changes in the amplitude of resting state oscillations, we investigated the dynamics of this simplified oscillator (see Materials and methods) in the presence of periodic forcing (i.e. stimulation) and various levels of noise. In can be shown that as input noise increases during the task state, limit cycle solutions are destabilized through a supercritical Hopf bifurcation. Analyzing the response of this simplified model both above and below the bifurcation threshold, that is, in the vicinity of the point where self-sustained (i.e. intrinsic) oscillations become unstable, we found that limit cycle solutions of this system are entrained by periodic forcing, although their amplitudes are highly sensitive to noise intensity. To see this, we computed resonance curves (*Figure 7B*) for this oscillator and compared the results for both high and low values of input noise variance. This simple qualitative analysis shows that the linearization induced by noise decreases the amplitude of the resonant solutions and increases the amplitude of all other forcing frequencies by transitioning through a critical state. Spectral clustering, in which the power and/or amplitude of forced solutions are concentrated near the intrinsic resonance, is a state of weak susceptibility to entrainment. The consequences of input-induced linearization in the simple conceptual model above are two-fold: (1) it causes a suppression of resonant oscillations, and (2) it augments the amplitude of non-resonant solutions. This was found to be in agreement with the state-dependency observed in our full spiking model as depicted in *Figure 7C*.

## Discussion

Alpha activity has been implicated in a wide variety of physiological and cognitive functions (*Başar and Basar, 2012*; *Mierau et al., 2017*). Numerous recent studies have demonstrated that periodically modulated electric fields can interfere with alpha oscillations to trigger a measurable effect on stimulus perception (*Chanes et al., 2013*; *Cecere et al., 2015*) and task performance by entraining these rhythms (*Henry et al., 2014*; *Samaha and Postle, 2015*). However, the efficacy of these approaches has been shown to depend heavily on brain states and to fluctuate according to internally governed dynamics (*Zaehle et al., 2010*; *Miniussi et al., 2013*; *Ruhnau et al., 2016*; *Alagapan et al., 2016*).

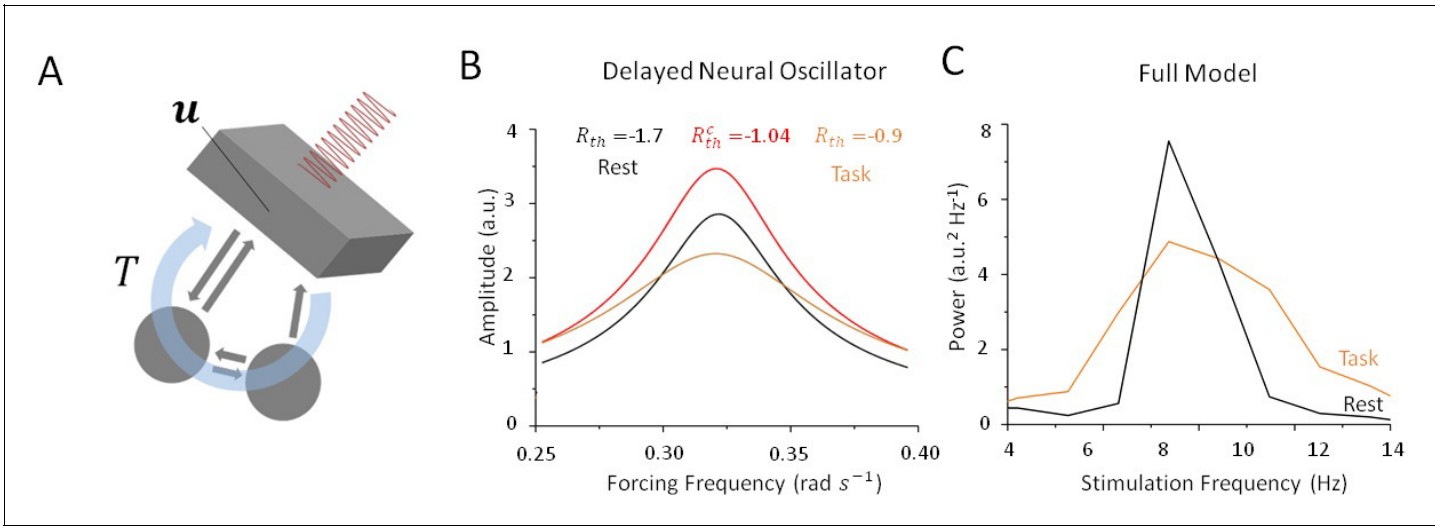

**Figure 7.** Stochastic resonance in a noisy and delayed neural oscillator. **(A)** We investigated the dynamics of a conceptual delayed feedback model in the presence of periodic forcing (i.e. stimulation) as a proxy for the thalamo-cortical circuit subjected to additive noise. The mean activity $u$ experiences delayed feedback with delay $T$, additive noise and periodic forcing. **(B)** Amplitude of entrained solution both above and below the Hopf bifurcation threshold. Below the bifurcation (black line), the linear gain $|R_{th}|$ remains high because noise intensity is small. Solutions have a high amplitude only near the intrinsic critical frequency. Above the bifurcation (orange line), the linear gain $|R_{th}|$ becomes smaller under the effect of noise. As a consequence, the amplitude of all non-resonant solutions increases, while the amplitude of resonant solution decreases. **(C)** Peak power at $\omega_{stim}$ in the rest (black) and task (orange) states as stimulation frequency varies in the vicinity of $\omega_o$. Although resulting from a more complex model, the effect remains qualitatively similar to the simplified case in (B).
DOI: https://doi.org/10.7554/eLife.32054.008

To better understand this phenomenon and to elucidate previous experimental findings (*Alagapan et al., 2016*), we here combined computational and mathematical approaches and analyzed a computational model of the thalamo-cortical system exhibiting resting state alpha oscillations. The dynamics of our network model were analyzed in two conditions: the rest state, which we have defined as a regime of weak afferent input to the thalamus, and the task-engaged state, where thalamic drive is high to reflect experimentally imposed conditions. We found that a destabilization of alpha oscillations occurred during the transition between rest and task-engaged states, in which both cortical and sub-cortical spiking patterns switched from correlated and synchronous to irregular and asynchronous. Then, applying periodic stimulation to cortical neurons, we found that the susceptibility of the system to external perturbations was strongly controlled by the presence (or absence) of an oscillatory attractor (i.e. the alpha oscillation) and as such depended heavily on state. By comparing the stimulation waveform with the cortical firing-rate responses for weak stimulation amplitudes, we found that SR was involved in improving cortical susceptibility to entrainment. While in the rest state, applied stimulation had a weak impact on the activity of cortical neurons whose dynamics remained locked to internal recurrent oscillations. Arnold tongues (i.e. regions in the stimulation parameter space where network activity is phase locked to the stimulation) were found to be very narrow. By contrast, in the task-engaged state, excitatory and inhibitory cortical neurons were freed from the intrinsic attractor and found to be more susceptible to entrainment, as shown by significantly larger Arnold tongues. Using a simplified neural oscillator model, our mathematical analysis revealed that thalamic input to the cortex, playing a similar role as noise, could control the stability of intrinsic oscillations by provoking a bifurcation, and by doing so, could amplify the response to non-resonant stimuli. Aside from the state-dependent input and/or noise, only stimulation parameters, such as amplitude and frequency, were changed in our study. Taken together, our results show that internal control over attractor states shapes the susceptibility of cortical populations to stimulation and entrainment. Our modelling results further suggest that the thalamo-cortical system behaves like a non-linear oscillator whose susceptibility to periodic forcing is gated by noise through SR and a passage through a bifurcation.

SR is widely appreciated as a mechanism that regulates cortical excitability (*Miniussi et al., 2013*) and has measurable effects on cognitive performance and perception (*van der Groen and Wenderoth, 2016*). Our computational results demonstrate the important role of SR, and thus of asynchronous and irregular neural dynamics, in regulating the responses of cortical neurons to stimulation. Using a simplified neural oscillator model, we have shown that noise — playing the role of uncorrelated inputs from the thalamus — can suppress oscillation through a Hopf bifurcation. Practically speaking, this is accomplished by a noise-induced linearization of the response function of the system, a generic effect observed in systems with threshold non-linearities (see *McDonnell and Abbott, 2009*) and references therein), which can further be shown to impact the frequency and amplitude of oscillatory solutions in non-linear systems (*Lefebvre et al., 2015*; *Hutt et al., 2016*). Such linearization also plays a fundamental role in dithering, a mechanism by which noise is added to a signal, prior to its digitalization, to increase its integrity (*McDonnell and Abbott, 2009*). Formulated in the context of neural systems, irregular and/or uncorrelated fluctuations can thus be used to increase the saliency or fidelity of neural responses. Interestingly, this 'correlation-induced blockade' has been previously discussed with respect to alpha oscillations (*Klimesch et al., 2007*), where large-scale synchrony has been hypothesized to implement a functional inhibition gate through which increased synchronized fluctuations shut down the mutual coding capacity of neural networks (*Zohary et al., 1994*; *Greenberg et al., 2008*). In this perspective, we hypothesize that alpha oscillations — as well as other intrinsic attractors — block perturbations of neural activity, triggered either by endogenous (e.g. inter-area signaling) or exogenous (e.g. stimulation) manipulations. Future work will show whether endogenous correlated neural activity, which goes beyond the purely rhythmic activity observed here, could also play a role in regulating susceptibility to entrainment, and more generally in control.

We here note that, in line with previous theoretical (*Hutt et al., 2016*) and experimental (*Mierau et al., 2017*) findings, the peak alpha frequency was found to change as a function of thalamic drive between 8 and 8.5 Hz. This can be seen for instance in *Figure 3C* (top), *Figure 5A,B* and *Figure 6A,B*, where the peak frequency is slightly higher in the task-engaged state than in the rest state. This is because inputs from the thalamus to the cortex, whose variance changes as a function of thalamic drive, correspond to a stream of irregular fluctuations (i.e. Poisson noise) that are known

to alter the oscillatory properties of non-linear networks (*Lefebvre et al., 2015*). As here these fluctuations possess stationary statistics (i.e. mean and variance that did not fluctuate in time), no temporal variability was observed in the peak frequency of endogenous oscillations. As such, this non-linear phenomenon, which is well understood mathematically, does not impact our conclusions.

Our results are conceptually connected to experimental findings in studies of tACS. In the task-engaged state, successful entrainment was reflected in a shift of the oscillation frequency to the stimulation frequency. This agrees with the findings of multiple studies of tACS that point towards a successful shift in the alpha peak frequency to the stimulation frequency during stimulation (e.g. *Helfrich et al., 2014*). Several studies using tACS (see *Thut et al., 2017* and references therein) and a recent study (*Minami and Amano, 2017*) that uses amplitude-modulated tACS (AM-tACS) have employed some type of visual task, ranging from a simple vigilance task to prevent participants from falling asleep all the way to sophisticated visual psychophysics paradigms.

As for any modeling work, a key aspect to discuss is the level of abstraction. The electrical field applied by tACS is the strongest in the cortex due to its closest proximity to the stimulation electrodes. However, any modulation of cortical network dynamics by stimulation must, almost by definition, also alter activity in the regions interconnected with the targeted cortical circuits, including the thalamus. However, dissecting the effects of stimulation on an interconnected network remains a challenge because of the feedback nature of the network dynamics. Here, we have chosen to study a reduced, relatively abstract model of the thalamo-cortical system. By contrast, in a recent study in which we focused on the cellular and synaptic biophysics and their role in shaping the response to stimulation, we examined the response to stimulation of thalamus in isolation (*Li et al., 2017*). Qualitatively, the findings from the two studies agree as the detailed thalamic model suggested that faster oscillations, typically associated with the task-engaged state, are more responsive to stimulation than the task-negative alpha oscillations.

Our results further raise interesting questions regarding sensory entrainment paradigms (*Keitel et al., 2017*). In the presence of periodic sensory stimulation, or sensory inputs possessing a rhythmic structure, thalamic drive would also possess a rhythmic structure. It would prove interesting to see how these fluctuations in thalamo-cortical inputs would interact with endogenous oscillations, and whether brain state fluctuations could gate their propagation towards cortical networks. Given the central role of thalamus in sensory gating (e.g. *Sherman, 2007*; *Poulet et al., 2012*), sensory rhythmic inputs that reach the thalamus first may have a different effect on thalamo-cortical oscillations than does cortical stimulation. Yet another fundamental difference is the magnitude of the input. Sensory input to the thalamus (e.g., retinal input to the LGN) is strong enough to cause spiking in the thalamus, which is time-locked to the temporal structure of the visual input (*Hirsch et al., 2015*). Thus such input is super-threshold in comparison to the subthreshold stimulation examined here.

In summary, our results demonstrate that target engagement by stimulation is state-dependent. The fact that the presence of a pronounced endogenous oscillation limits the impact of stimulation in terms of entrainment points towards important questions for the use of such stimulation paradigms in research, and ultimately in the clinic. First, the overall state of the network needs to be measured and documented in studies that investigate brain stimulation for the modulation of brain rhythms. Second, it remains mostly unclear how adjusting the frequency of the alpha oscillation affects cognition and behavior (*Romei et al., 2016*; *Mierau et al., 2017*). Yet, a reduced alpha peak frequency is a hallmark of a range of neurological and psychiatric illnesses (e.g. *Rossini et al., 2007*) and probably an overall marker of brain health (*Klimesch, 1999, 2012*). In such work, our findings suggest that states of low alpha power represent the ideal state for the application of stimulation, and that modulation of brain oscillations is best achieved in states of low endogenous alpha rhythmic activity. This finding may be important for the next generation of brain stimulation paradigms developed using rational design.

## Materials and methods

### Model

We have developed and analyzed a model of the thalamo-cortical system in which recurrent interactions between the different neural populations generate strong synchronous activity within the alpha

band (8 Hz–12 Hz). An illustration of the model structure and connections is presented in *Figure 1A*. The cortical population is composed of recurrently connected excitatory pyramidal neurons and inhibitory interneurons, and both interact with thalamic and reticular populations via delayed connections. The spiking activity of all neurons is modeled by a non-homogenous Poisson process,

$$X_n^j(t) \rightarrow Poisson\left(f\left[u_n^j(t)\right]\right)$$

where $X_n^j(t) = \{t_l\}\sum \delta_n^j(t-t_l)$ is the spike train of the $j^{th}$ neuron in the population $n = \{e, i, \mathrm{LGN}, \mathrm{RTN}\}$. The activation function $f[u] = f_o(1+\exp\{-\beta(u-h)\})^{-1}$ represents a saturating firing-rate function that relates monotonically to the cellular membrane potential $u$, with gain $\beta$, maximal rate $f_o$ and threshold $h$. The mean somatic membrane potentials $u_n^j$ of all neurons in the network obey the set of dynamic equations

$$\alpha_n^{-1}\frac{du_n^j(t)}{dt} = -u_n^j(t) + b\,v_n^j(t) + \sum_m S_{nm}(t) + I_n + \sqrt{2D_n}\,\xi_n^j(t) + S_{e,i}(t)$$

where we have spike frequency adaptation

$$a_n^{-1}\frac{dv_n^j(t)}{dt} = -v_n^j(t) + u_n^j(t)$$

and where recurrent inputs are

$$S_{nm}(t) = N_n^{-1}\sum_{k=1}^{N_n} W_{nm}^{jk}\cdot E_n^k\left(t-\tau^{jk}\right)\,.$$

The rates $\alpha_n$ and $a_n$ define the time scale of the somatic membranes and adaptation, respectively. The post-synaptic potentials $E_n^k(t)$ are computed by convolving spike trains with exponential synapses with time constant $\tau_m$ that is,

$$E_n^k(t) = \int_o^t X_n^k(s)\,\frac{1}{\tau_m}e^{-(t-s)/\tau_m}\,ds;$$

All populations, including the sub-cortical neurons (relay, reticular), are mutually coupled with sparse and spatially topographic projections (*Hellwig, 2000*). The synaptic connectivity kernels between neurons of index $j$ and $k$, respectively from population $m$ and $n$ are given by Gaussians

$$W_{nm}^{jk}(c) = \frac{w_{nm}^o(c)}{\sqrt{2\,\pi\sigma_{m,n}^2}}\exp\left[-\frac{1}{2\,\sigma_{n,m}^2}(x(j)-x(k))^2\right]$$

with connection probability $c = 0.2$ and $\sigma_{m,n}^2$ as the spatial spread of the inter-neuronal connections. The network is also subjected to propagation delays $\tau^{jk} = |x(k) - x(j)|\,v^{-1}$ due to finite axonal conduction velocity $v$=0.35 m/s (*Hutt et al., 2003*). Furthermore, a delay of $\tau_{th}$ =45 ms was included between thalamus (RTN, LGN) and cortex $(e, i)$, and a delay of $\tau_{rtn}$=10 ms between reticular (RTN) and relay (LGN) populations.

## Thalamic drive and brain states

In addition to fixed bias inputs, $I_n$, neurons in the network are subjected to Gaussian white noise, $\xi_n$, of intensity, $D_n$. To represent an increase in sensory afference to the thalamus in two distinct conditions, the intensity of the noise driving the lateral geniculate nucleus (LGN) neurons is increased. To mimic externally imposed experimental states, we analyzed the dynamics of this network model in two conditions:

**rest state:** regime of low thalamic drive (i.e. $D_{LGN} = 1\ \times 10^{-4}$);
**task-engaged state:** regime of high thalamic drive (i.e. $D_{LGN} = 1$).

Specifically, a transition between the rest and task-engaged states occurs whenever noise intensity at the LGN increases. Aside from this input, which was changed to set the system in the rest and/or task-engaged state, only the periodic stimulation parameters, such as amplitude and

frequency, were changed. We chose not to define the thalamic drive intensity for the task state as the optimal value found through SR (i.e. *Figure 3D*), because this value was found to be sensitive to the stimulation amplitude and frequency, which will vary in the subsequent analysis. To represent the effect of periodic stimulation, all excitatory and inhibitory cortical neurons were equally driven by a periodic input with waveform $S_{e,i}(t) = S \sin(2 \pi \omega_{stim} t)$, where $S$ is defined as the stimulation amplitude and $\omega_{stim}$ is the stimulation frequency. No stimulation is present for sub-cortical populations as non-invasive brain stimulation predominantly targets cortex.

For minimal thalamic drive (i.e. $D_{LGN} = 1 \times 10^{-4}$), the network engages intense resting state alpha activity at a frequency of about 8 Hz due to the combined action of slow spike-frequency-dependent adaptation and finite conduction velocity — resulting in propagation delays that enhance the prevalence of correlated rhythmic states (*Lefebvre et al., 2011*; *Deco et al., 2009*). The stability of the resulting oscillatory solutions is maintained and amplified due to thalamo-cortical feedback, leading to phase-locked dynamics across all neural populations involved, both cortical and sub-cortical. The activity of each population in the resting state is depicted in *Figure 1B*. The parameter values that were chosen were aligned within the physiological range in the literature and are summarized in *Table 1*. The response of the sub-cortical populations (i.e. LGN, RTN) can be seen to adopt a stable phase offset with respect to cortical activity. Such phase differences have been observed experimentally and shown to be sensitive to ongoing oscillatory state (*Slézia et al., 2011*). In our model, this phase lag occurs because of the propagation delay in the transition to and back from the cortex. When thalamic drive is increased (i.e. $D_{LGN} = 1$), global oscillations are suppressed through a noise-induced Hopf bifurcation (see below). Through this transition, noise triggers a gradual change in effective gain between populations, suppressing global oscillations and replacing them by asynchronous activity.

In addition, we mention that the unit of the model voltage and the model firing threshold is arbitrary, which reflects an invariance with respective to unknown physiological parameters. Consequently, it is difficult to compare experimental stimulation parameters, such as stimulus amplitude, to model parameters. One typical distinction in the literature is the differentiation between 'sub-threshold' and 'super-threshold' stimulation (in terms of generating an action potential), which conveys the notion of 'weak' versus 'strong' input. However, the stimulation amplitude alone does not determine in isolation whether an action potential is generated. Rather the electrical state of the neuron (in particular, its level of depolarization) plays an equally important role. Thus such classification makes most sense when applied to stimulation occurring when the neuron is quiescent and thus at the resting membrane voltage. In our work here, we do not directly model the membrane voltage but rather how spiking activity depends on input, which implies an underlying subthreshold depolarization. However, as a reference for discussion of our results in the context of different human non-invasive brain stimulation paradigms, the shape of the transfer function of input to spiking output in our model supports the conclusion that the range of amplitudes considered here would mostly fall into the 'weak' (subthreshold) category.

## Simulated EEG

In our model, encephalographic (EEG) dynamics are modelled by a weighted sum over somatic excitatory and inhibitory potentials, that is

$$A(t) = \frac{1}{N_e} \sum_{k=1}^{N_e} \phi_e^k u_e^k(t) + \frac{1}{N_i} \sum_{k=1}^{N_i} \phi_i^k u_i^k(t),$$

where $\phi_{e,i}^k$ are real positive coefficients. Here we assume that the network's fine-scale structure is unknown, and thus consider random weights $\phi_{e,i}^k = [0,1]$. However, we note that specific choices of the $\phi_{e,i}^k$ distributions can be made to increase the similarity of the network activity to physiological signals, such as LFPs and EEG (*Linden et al., 2010*, *Mazzoni et al., 2015*).

## Spectral analysis and phase distributions

Spectral analysis was performed using a fast Fourier transform (FFT) routine using freely available C++ scripts (*Press et al., 2007*). To quantify entrainment, we used a windowed discrete Fourier transform to compute the phase of the firing-rate activity for cortical excitatory cells across independent

**Table 1.** Model parameter efinitions and Values.

| Symbol | Definition | Value |
| --- | --- | --- |
| $\Omega$ | Network spatial extent | 1 |
| $N_e$ | Number of excitatory cells | 800 |
| $N_i$ | Number of inhibitory cells | 200 |
| $N_{th}$ | Number of thalamic cells | 200 |
| $N_{rtn}$ | Number of reticular cells | 200 |
| $\beta$ | Neural response function gain | 150 |
| $h$ | Firing rate threshold | 0.1 |
| $f_o$ | Maximal firing rate | 0.2 |
| $\tau_m$ | Membrane time constant | 1 |
| $\alpha_e$ | Membrane rate constant — excitatory cortical cells | 0.9 |
| $\alpha_i$ | Membrane rate constant — inhibitory cortical cells | 1.3 |
| $\alpha_{th}$ | Membrane rate constant — thalamic neurons | 0.5 |
| $\alpha_{rtn}$ | Membrane rate constant — reticular cells | 0.5 |
| $a$ | Neural adaptation rate constant | 0.01 |
| $b$ | Neural adaptation gain | 0.3 |
| $I_e$ | Constant current bias | 0 |
| $I_i$ | Constant current bias | −0.3 |
| $I_{th}$ | Constant current bias | −0.3 |
| $I_{rtn}$ | Constant current bias | −0.3 |
| $\tau_{th}$ | Thalamo-cortical delay | 45 ms |
| $\tau_{rtn}$ | Reticular-thalamic delay | 10 ms |
| $c$ | Conduction velocity | 0.35 m/s |
| $\rho$ | Connection probability | 0.2 |
| $w^o_{e-}$ | Synaptic connection strength | 20.4 |
| $w^o_{e-}$ | Synaptic connection strength | 30.6 |
| $w^o_{i-}$ | Synaptic connection strength | −30.6 |
| $w^o_{i-}$ | Synaptic connection strength | 20.4 |
| $w^o_{e-}$ | Synaptic connection strength | 34 |
| $w^o_{e-}$ | Synaptic connection strength | 34 |
| $w^o_{lgn-}$ | Synaptic connection strength | 85 |
| $w^o_{lgn-}$ | Synaptic connection strength | 85 |
| $w^o_{lgn-}$ | Synaptic connection strength | 34 |
| $w^o_{rtn-}$ | Synaptic connection strength | −34 |
| $\sigma^2_{e-}$ | Synaptic connection range | 0.01 |
| $\sigma^2_{e-}$ | Synaptic connection range | 0.01 |
| $\sigma^2_{i-}$ | Synaptic connection range | 0.25 |
| $\sigma^2_{i-}$ | Synaptic connection range | 0.25 |
| $\sigma^2_{e-}$ | Synaptic connection range | 0.01 |
| $\sigma^2_{e-}$ | Synaptic connection range | 0.01 |
| $\sigma^2_{th-}$ | Synaptic connection range | 0.25 |
| $\sigma^2_{th-}$ | Synaptic connection range | 0.25 |
| $\sigma^2_{th-}$ | Synaptic connection range | 0.25 |
| $\sigma^2_{rtn-}$ | Synaptic connection range | 0.25 |
| dt | Integration time step | 0.1 |

DOI: https://doi.org/10.7554/eLife.32054.009

trials, and to this phase with the phase of the stimulation applied. As such, we computed the response of the network in rest and task conditions to stimulation delivered with a random phase for 200 independent trials of 2 s. In each trial, we randomly selected a time window of 500 ms and computed the phase difference, at the stimulation frequency of 11 Hz, between firing-rate response and input stimuli. We did this to ensure no phase consistency in the endogenous network response between successive trials on which we computed phase differences. We then evaluated the distribution of these phase differences across all trials in both conditions. Results are shown in *Figure 3E and F*. In each trial, cortical neurons were driven with a 11 Hz stimulus applied at a random phase. To ensure statistical significance, we successively shuffled and separated trials into two groups and computed the distribution of phase differences in each case. We then computed the circular variance of these distributions as a measure of phase clustering (i.e. small circular variance indicating more clustered phases around the mean). We then computed the probability of observing the variance in the unshuffled data.

## Mutual information

We computed mutual information between the stimulation signal $S(t)$ and the mean firing-rate response $r(t)$ to measure how well the stimulation waveform was reflected in the spiking patterns of cortical neurons across independent trials. We assumed that for sufficiently high firing rates, the random fluctuations impacting the network responses can be approximated by Gaussian white noise, and computed,

$$MI = \frac{1}{2}log_2[1 + SNR]$$

where $SNR = \frac{\rho}{1-\rho}$ is the signal to noise ratio and $\rho$ is the crossspectral correlation defined by

$$\rho = \frac{\Phi_{rS}(0)}{\sqrt{\Phi_{rr}(0)\Phi_{SS}(0)}}$$

The convolutions $\Phi_{xy}(f) = \int_0^\infty d\omega\, \tilde{x}(\omega)\tilde{y}(\omega - f)$ measure the covariance between the Fourier transforms $\tilde{x}(f)$ and $\tilde{y}(f)$ of two time series $x(t)$ and $y(t)$. We computed the mutual information between input signal $S(t)$ and responses $r(t)$ for increasing noise variance $D$ across all populations of the network. We did this to confirm the presence of stochastic resonance, where irregular fluctuations first improve and then hinder the detection of weak signals, and also to compensate for firing-rate saturation (which bounds the variance of input- and noise-induced fluctuations).

## Noise-induced transition and resonance curves for a simplified delayed neural oscillator

The thalamo-cortical system has been thoroughly studied, both in experiments and in models, and its low frequency dynamics have been shown to be largely determined by the presence of a delayed feedback loop between cortex and thalamus (e.g. *Roberts and Robinson, 2008*). A thorough derivation of the reduced dynamics for the detailed spiking model that we used in the simulations is far beyond the scope of this paper, and is highly challenging due to the combined presence of sparse and topographic synaptic projections, multiple cell types and spiking activity. To better understand the mechanism involved in shaping resting-state cortical oscillations, we built a simplified conceptual neural oscillator model, whose limit cycle solutions emerge due to the combined presence of delayed feedback and slow spike-frequency adaptation,

$$\frac{dU}{dt} = -U + bV + F_{th}[U(t-T)] + S(\mathrm{t})$$

$$s\frac{dV}{dt} = -V + U$$

with feedback response function $F_{th}(U) = \frac{g}{2}\left(1 + \mathrm{erf}\left[\frac{U}{\sqrt{2\Gamma}}\right]\right)$ for some gain g<0 and time delay $T$. The

membrane potential proxy $U$ here denotes the mean somatic membrane potential of cortical neurons, subjected to re-entrant inputs back from the thalamus. The response function also depends on the state-dependent noise variance $\Gamma$. This results from non-linear interactions between noise and a sigmoidal non-linearity, which is revealed by performing a mean-field reduction (*Lefebvre et al., 2015*; *Hutt et al., 2016*). Note that a similar model has been used before to study state-dependent entrainment and outlasting effects observed in experimental data in humans (*Alagapan et al., 2016*).

Assuming slow adaptation ($s$ large) and the temporal evolution close to the systems fixed point, the smoothing of the neurons' response functions (*Lefebvre et al., 2015*) leads to the effective linear dynamics with periodic forcing,

$$\frac{dU}{dt} = (b-1)U + R_{th}U(t-T) + S(\mathrm{t}) \tag{1}$$

with $R_{th} = \frac{g}{\sqrt{2\pi\Gamma}}\exp\left[-\frac{u_o^2}{2\Gamma}\right]$ and fixed point $U_o$. It has been shown that the value of the gain $R_{th}$ is inversely proportional to the intensity of noise in the system. Indeed, for low values of $\Gamma$ (i.e. in the rest state), $|R_{th}|$ remains high due to the steepness of the neuron response function. By contrast, as $\Gamma$ increases (i.e. in the task state), $|R_{th}|$ decreases (Hutt et al. 2016). The impact of state-dependent noise on oscillatory solutions can thus be quantified by analyzing how the stability of the system depends on the parameter $\Gamma$. It can be shown that increasing the noise variance in the system above causes a destabilization of limit cycle solutions through a supercritical Hopf bifurcation. Setting $S = 0$ and using the ansatz $U = U_o + \delta U\, e^{-\lambda t}$ with $\lambda = \pm i\,\omega_c$, where $\omega_c$ in the linearized system above, one obtains after separating real and imaginary parts,

$$T\sqrt{R_{th}^{c\,2} - 1} = \cos^{-1}\left(1/R_{th}^c\right)$$

Setting $T = 2 \cdot \tau_{th} = 90$ ms (i.e. twice the thalamo-cortical delay), one then obtains $R_{th}^c \approx -1.05$.

The derivation of the resonance curve for a linear delayed system with periodic forcing is easily accomplished by finding the amplitudes of its solution, which can be computed explicitly via substitution. Let us assume an entrained oscillatory solution of the form

$$U(t) = A(S,f)\sin(ft) + B(S,f)\cos(ft)$$

Substituting this ansatz in *Equation (1)* above and solving for the $A(S,f)$ and $B(S,f)$, one can then compute the amplitude of the solution $u$ as

$$\|U\| = \sqrt{A(S,f)^2 + B(S,f)^2},$$

where

$$A(S,f) = -S\frac{\cos(fT)R_{th} - 1}{\left(R_{th}^2 + 2\sin(fT)R_{th}T - 2\cos(fT)R_{th} + 1 + f^2\right)}$$

and

$$B(S,f) = -S\frac{\sin(fT)R_{th} + f}{\left(R_{th}^2 + 2\sin(fT)R_{th}T - 2\cos(2fT)R_{th} + 1 + f^2\right)}.$$

## Acknowledgements

This work has been supported by the Natural Sciences and Engineering Research Council of Canada (JL). Research reported in this publication was supported in part (FF) by the National Institute of Mental Health of the National Institutes of Health under Award Number R01MH111889. The content is solely the responsibility of the authors and does not necessarily represent the official views of the National Institutes of Health.

## Additional information

### Funding

| Funder | Grant reference number | Author |
|---|---|---|
| Natural Sciences and Engineering Research Council of Canada | RGPIN-2017-06662 | Jérémie Lefebvre |
| National Institute of Mental Health | R01MH111889 | Flavio Frohlich |

The funders had no role in study design, data collection and interpretation, or the decision to submit the work for publication.

### Author contributions

Jérémie Lefebvre, Axel Hutt, Conceptualization, Resources, Data curation, Software, Formal analysis, Supervision, Funding acquisition, Validation, Investigation, Visualization, Methodology, Writing— original draft, Project administration, Writing—review and editing; Flavio Frohlich, Conceptualization, Resources, Supervision, Writing—original draft, Project administration, Writing—review and editing

### Author ORCIDs

Jérémie Lefebvre (iD) http://orcid.org/0000-0003-0369-4565
Axel Hutt (iD) http://orcid.org/0000-0003-0041-7431
Flavio Frohlich (iD) http://orcid.org/0000-0002-3724-5621

### Decision letter and Author response

Decision letter https://doi.org/10.7554/eLife.32054.011
Author response https://doi.org/10.7554/eLife.32054.012

## Additional files

### Supplementary files

• Transparent reporting form DOI:

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
