## [Decision Letter]

Thank you for submitting your article "Stochastic Resonance Mediates the State-Dependent Effect of Periodic Stimulation on Cortical Alpha Oscillations" for consideration by *eLife*. Your article has been reviewed by three peer reviewers, one of whom, Saskia Haegens, served as Guest Reviewing Editor, and the evaluation has been overseen by Sabine Kastner as the Senior Editor. The following individuals involved in review of your submission have agreed to reveal their identity: Caroline Di Bernardi Luft and Vincenzo Romei.

The reviewers have discussed the reviews with one another and the Reviewing Editor has drafted this decision to help you prepare a revised submission.

Summary:

This manuscript by Lefebvre et al. addresses how intrinsic oscillatory activity affects the brain's susceptibility to rhythmic transcranial stimulation, using a computational model of the thalamocortical system. Systematically manipulating stimulation frequency and amplitude, as well as the state of the network during stimulation, ranging from rest to active task engagement, the authors observed evidence for stochastic resonance. Cortical spiking responses were phase locked to the rhythmic stimulation only for intermediate values of thalamic drive; below this drive, endogenous network activity suppressed cortical entrainment, while above this value noise decreased the saliency of the evoked responses. The reviewers deem this paper interesting and relevant and believe it could be further strengthened by expanding on more realistic scenarios and physiological plausibility and referring directly to existing work empirically providing such links.

Essential revisions:

The reviewers raise a number of concerns that must be adequately addressed before the paper can be accepted. We suggest to expand the discussion of the model further by considering realistic (experimental) scenarios including rhythmic sensory stimulation and different neuromodulation protocols:

1) The authors should clarify the simulated stimulation intensity and how it compares to the stimulation intensities used in experimental studies with humans. It is not clear how strong the simulated stimulation was, which is key if we want to test this model's predictions using weak vs. strong currents (e.g. tACS vs. TMS). The authors must explicitly state whether the intensity was sub or supra-threshold and how it compares to the current reaching the brain in studies with humans.

2) Do the authors think the results presented here are exclusive to cortical stimulation approaches, or would this (potentially) also hold for "entrainment" type sensory stimulation paradigms? In that light, it has been shown recently (Keitel et al., 2017 NeuroImage) that the temporal structure of natural stimuli is hardly ever fully rhythmic but possesses certain rhythmic structures. Examining periodic brain responses elicited by strictly rhythmic stimulation might therefore represent ideal, yet isolated cases (neurostimulation being one of those). Therefore, a first question here is what kind of task state the system has been tested on? Have the authors thought of testing the equivalent of a rhythmic sensory input that could induce a rhythmic thalamic drive? How would this translate in terms of cortical oscillatory activity? The second question is whether a quasi-periodic stimulation (think about speech) would exert the same impact on the endogenous ongoing oscillatory activity in the computational model presented?

3) The discussion could be improved by using the findings to explain results from empirical work. For example:

– A study with intracranial recordings in humans (Ezzyat et al., Current Biology 2017) observed that the stimulation only boosted memory on trials in which the expected oscillatory pattern was not present, a finding which could be nicely explained by the current model. The authors might go a bit too far saying that stimulating at rest has no effect due to the strong endogenous frequency, as this is not entirely supported by their evidence (which seems to suggest that the stimulation frequency has to be much closer to the endogenous frequency than at higher thalamic drive). Possible effects on LTP and the aftereffects are also not taken into account by the model nor were they tested, so it would be good to see some discussion on how their findings connect to those, or to address it as a limitation.

– A recent MEG tACS paper by Minami and Amano, 2017 showed how stimulating at slower or faster alpha frequencies than the individual alpha via tACS directly modulates the individual alpha peak itself, which becomes aligned to the externally imposed rhythm. Interestingly, the authors test their model during a visual task (which is impacted by the tACS manipulation), thus perfectly in line with the model proposed here when looking at modulation of the intrinsic alpha peak during task.

4) The authors might want to incorporate the recently published paper by the senior author (Li, Henriquez, and Frohlich, 2017, Plos Comp. Biol.) into their Discussion (and Introduction). Although the study is different, the thalamic model somehow similar and the findings and ideas related. It would be good if the authors could point out the main differences between them and discuss their findings against those. Li et al. observed entrainment depending on the stimulated frequency (e.g. entrainment was stronger when stimulating at γ). These findings could be discussed in the context of whether stimulating at different frequencies would have a different impact on the observed effects.

Furthermore, there were several questions on what the full spectral output of the model looks like, whether this is physiologically plausible, and to what extent the results would hold for other (oscillatory) states:

5) It would be good if the authors could clarify how much variability is in the model and the outputs of stimulation simulations, including in terms of the model's temporal and inter-trial variability, which is especially important considering the non-stationary nature brain signals even under low thalamic drive. It would be good to see measures of variability in order to understand the stability of the responses and how this might affect the outcomes of the stimulation, e.g., does the model result in a constant endogenous 8Hz oscillatory output? Or is there some variation around this central frequency of 8Hz? And if so, how is the model assuming a transition from 8Hz to 11Hz? Would this be physiologically plausible?

6) Figure 3 shows a nice representation of the spectrum during the stimulation at 11 Hz over different levels of thalamic drive. It would be good to see the same figure without the stimulation in order to have an overview of the changes in the spectrum caused by the thalamic drive alone. Additionally, it would be good to see what the EEG spectrum looks like under these different levels of thalamic drive as in rest and task (in Figure 4, without stimulation).

7) "Overall, our findings suggest that modulating brain oscillations is best achieved in states of low endogenous rhythmic activity," Shouldn't that say, "in states of low endogenous alpha activity"? Or do the authors think this holds for other frequencies as well? And if so, which and why, and where is the evidence for such a general claim?

8) It would be helpful to the reader if the authors could clearly define what they mean by "state", e.g.:

– Results section: "or other sensory inputs meant to fluctuate as a function of state"

– Figure 1 legend: "Sensory inputs, whose intensity scales with state"

Also later when discussing "state-dependent effects", use of the term "state" is ambiguous. Do the authors exclusively mean externally imposed experimental state (rest vs task condition)? Or do they also consider intrinsic (spontaneous?) brain state? Or both? Especially considering these two are not independent (e.g., spontaneous state might affect the response to external input), this should be discussed. In that light: "Visual inputs (or other sensory inputs meant to fluctuate as a function of state) are sent through the LGN to cortical areas for further processing". Do the authors mean to imply that thalamic input is unmodulated? i.e., does LGN, in this model, simply relay visual input? Or is LGN activity also modulated based on intrinsic brain state? All of this should be unambiguously spelled out in the manuscript. It would also be helpful to have the operational definition of task and rest states ("the rest state – which we have defined as a regime of weak inputs to the thalamus, and the task state – where thalamic drive is high") in the Introduction or early in the Results section.

9) In addition to a qualitative representation, a quantitative approach specifying the extent to which the proposed model is statistically robust could make it a much stronger contribution. The authors should substantiate their claims, e.g., focusing on the results presented in Figure 3: for which thalamic drive values is spike rate correlation significantly different for stimulation vs sham? Similarly, when is 8-Hz power sig different from 11-Hz power? Same for the phase-locking analysis. Figure 3 shows an asterisk for the peak of mutual information, is that signifying any statistical test result? All of this should be quantified using appropriate statistics, and documented in the Results section. (Same applies to results presented in the following figures.)

---

## [Author Response]

Essential revisions:The reviewers raise a number of concerns that must be adequately addressed before the paper can be accepted. We suggest to expand the discussion of the model further by considering realistic (experimental) scenarios including rhythmic sensory stimulation and different neuromodulation protocols:1) The authors should clarify the simulated stimulation intensity and how it compares to the stimulation intensities used in experimental studies with humans. It is not clear how strong the simulated stimulation was, which is key if we want to test this model's predictions using weak vs. strong currents (e.g. tACS vs. TMS). The authors must explicitly state whether the intensity was sub or supra-threshold and how it compares to the current reaching the brain in studies with humans.

This is an important point, thank you for drawing our attention to it. The concept of subthreshold and superthreshold stimulation is a helpful concept to illustrate the relative strength of stimulation. What often gets ignored is the fact that the classification for sub- and superthreshold makes most sense for a neuron which is absolutely quiet and exhibits a membrane voltage close to the resting membrane voltage. However, if a neuron is active, even a very weak perturbation can “lift” it above the threshold for action potential generation. It is less clear how to call such a perturbation. Technically it causes a spike and is thus super-threshold but then according to the more traditional definitions this is a very weak perturbation which would barely alter the membrane voltage in a quiet cell and thus clearly a subthreshold input. We decide here to adopt the classical definitions and thus label our stimulation as subthreshold. Note that due to the mathematical nature of our model, this is no threshold as such. However. neuronal activity is governed by Poisson processes and stimulation-induced fluctuations modulate firing probability. The stimulation intensities we used generally were not sufficiently strong to in isolation cause a neuron to fire since it did not make the neuron to reach the threshold of the firing rate response 𝑓[𝑢]. Further, and because our model is non-dimensional (now explicitly said in the Materials and methods section), the units for stimulation intensity are arbitrary (au). To clarify these points, we have now added the following in the model description:

“In addition, we mention that the unit of the model voltage and the model firing threshold is arbitrary which reflects an invariance with respective to unknown physiological parameters. […] However, as a reference for discussion of our results in the context of different human non-invasive brain stimulation paradigms, the shape of the transfer function of input to spiking output in our model supports that the range of amplitudes considered here mostly would fall into the “weak” (subthreshold) category.”

2) Do the authors think the results presented here are exclusive to cortical stimulation approaches, or would this (potentially) also hold for "entrainment" type sensory stimulation paradigms? In that light, it has been shown recently (Keitel et al., 2017 NeuroImage) that the temporal structure of natural stimuli is hardly ever fully rhythmic but possesses certain rhythmic structures. Examining periodic brain responses elicited by strictly rhythmic stimulation might therefore represent ideal, yet isolated cases (neurostimulation being one of those). Therefore, a first question here is what kind of task state the system has been tested on? Have the authors thought of testing the equivalent of a rhythmic sensory input that could induce a rhythmic thalamic drive? How would this translate in terms of cortical oscillatory activity? The second question is whether a quasi-periodic stimulation (think about speech) would exert the same impact on the endogenous ongoing oscillatory activity in the computational model presented?

We thank the referee for this relevant comment. We have not examined this possibility and have considered thalamic drive (i.e. “sensory input”) as a noisy process with stationary statistics. Sensory entrainment possessing a rhythmic structure would certainly interact with cortical oscillations, but it is difficult to evaluate this scenario with the current model given that the thalamic drive is – by itself – the main driver of changes in brain state. A model comprising multiple (and possibly self-driven or attentional) input sources could potentially answer this question. We can hypothesize, on the basis of our results, that “brain state fluctuations” would gate the influence of sensory fluctuations in a similar way. To take into account this possibility, we have added the following in the Discussion:

“Our results further raise interesting questions regarding sensory entrainment paradigms (Keitel et al., 2017). […]Thus such input is superthreshold in comparison to the subthreshold stimulation examined here.”

3) The discussion could be improved by using the findings to explain results from empirical work. For example:– A study with intracranial recordings in humans (Ezzyat et al., Current Biology 2017) observed that the stimulation only boosted memory on trials in which the expected oscillatory pattern was not present, a finding which could be nicely explained by the current model. The authors might go a bit too far saying that stimulating at rest has no effect due to the strong endogenous frequency, as this is not entirely supported by their evidence (which seems to suggest that the stimulation frequency has to be much closer to the endogenous frequency than at higher thalamic drive). Possible effects on LTP and the aftereffects are also not taken into account by the model nor were they tested, so it would be good to see some discussion on how their findings connect to those, or to address it as a limitation.

We thank the reviewer for pointing us to the exciting work about the stimulation effect on memory performance. Indeed the boost of low-encoding memory states by stimulation seems to be linked to our own results. For instance, Figure 5 – task state, shows that stimulation with high frequency diminishes activity at endogenous frequency and enhances activity at stimulation non-endogenous frequency. We have added the given reference to the Introduction as an additional important application of brain stimulation.

The reviewer is right in stating that detailed single-neuron properties such as LTP are not considered. The present study considers rate-coding neural population dynamics in order to be able to extract mathematical relations explaining the underlying mechanisms of noisy thalamic driving. This mechanism is the switch between a stable limit cycle in the rest state to a stable focus in the task state. This insight allows understanding the impact of external stimuli. Future work surely will take into account more microscopic level properties, such as LTP or Ca^2+^+ – dynamics.

– A recent MEG tACS paper by Minami and Amano, 2017 showed how stimulating at slower or faster alpha frequencies than the individual alpha via tACS directly modulates the individual alpha peak itself, which becomes aligned to the externally imposed rhythm. Interestingly, the authors test their model during a visual task (which is impacted by the tACS manipulation), thus perfectly in line with the model proposed here when looking at modulation of the intrinsic alpha peak during task.

Thank you for mentioning this fascinating paper. Indeed, this study points towards a successful modulation of the alpha oscillation with tACS during a visual task. In addition, there are some important differences to be considered, perhaps most importantly that the authors of that study used a new form of tACS, termed amplitude-modulated tACS (AM-tACS), for which the mechanism of target engagement remains to be studied. Here is how we have updated the manuscript:

“Multiple studies are pointing towards a successful shift in the alpha peak frequency to the stimulation frequency during stimulation (e.g. Helfrich et al., 2014). Several studies using tACS (see Thut et al., 2017 and references therein) and a recent study (Minami and Amano, 2017) that uses amplitude-modulated tACS (AM-tACS) have employed some type of visual task, ranging from a simple vigilance task to prevent participants form falling asleep all the way to sophisticated visual psychophysics paradigms.”

4) The authors might want to incorporate the recently published paper by the senior author (Li, Henriquez, and Frohlich, 2017, Plos Comp. Biol.) into their Discussion (and Introduction). Although the study is different, the thalamic model somehow similar and the findings and ideas related. It would be good if the authors could point out the main differences between them and discuss their findings against those. Li et al. observed entrainment depending on the stimulated frequency (e.g. entrainment was stronger when stimulating at γ). These findings could be discussed in the context of whether stimulating at different frequencies would have a different impact on the observed effects.

Thank you for giving us the opportunity to comparatively discuss our studies. We have now added the following statement to the Discussion:

“The electrical field applied by tACS is the strongest in cortex due to closest proximity to the stimulation electrodes. […] Qualitatively, the findings from the two studies agree since the detailed thalamic model suggested that faster oscillations, typically associated with task-engaged state, are more responsive to stimulation than the task-negative, alpha oscillations.”

Furthermore, there were several questions on what the full spectral output of the model looks like, whether this is physiologically plausible, and to what extent the results would hold for other (oscillatory) states:5) It would be good if the authors could clarify how much variability is in the model and the outputs of stimulation simulations, including in terms of the model's temporal and inter-trial variability, which is especially important considering the non-stationary nature brain signals even under low thalamic drive. It would be good to see measures of variability in order to understand the stability of the responses and how this might affect the outcomes of the stimulation, e.g., does the model result in a constant endogenous 8Hz oscillatory output? Or is there some variation around this central frequency of 8Hz? And if so, how is the model assuming a transition from 8Hz to 11Hz? Would this be physiologically plausible?

We feel there are two aspects to this comment. The first, which concerns the trial-to-trial variability observed in the dynamics, peak frequencies and associated statistics, has been addressed in comment 9), below.

Second, we do expect – and have seen – variability in the peak frequency. This variability is however not temporal or from trial-to trial but is instead due to noise. Previous work from the authors has focused on this phenomenon theoretically (Lefebvre et al., 2015), numerically in presence of pulsatile stimulation (Herrmann et al., 2016) and also experimentally (Mierau et al., 2017). This jittering can be observed in Figure 3 (top), Figure 5 and Figure 6, where the peak frequency (outside the Arnold tongues) is slightly higher in the task vs rest state. However, we note that the acceleration due to increased thalamic drive is small (for the set of parameters chosen and due to the presence of thalamic population that stabilize the endogenous oscillation). Thalamic drive being modeled as a noisy process with stationary statistics (mean and variance that do not fluctuate in time, in contrast with Lefebvre et al., 2015), only small peak frequency shifts can be observed for the range of parameters explored. However, if the mean/variance of thalamic fluctuations was to fluctuate in time (using colored noise instead of GWN for instance or due to sensory entrainment (see comment 2) above), more significant changes in peak frequency could be observed, leading to a bending of the Arnold tongue (as seen in Herrmann et al., 2016 in Figure 4). As such, while we do observe this shift in our data, it does not impact our results.

We have clarified this point now in the manuscript by adding the following comment in the Discussion:

“We here note that, in line with previous theoretical (Hutt et al., 2016,) and experimental (Mierau et al., 2017) findings, the peak alpha frequency was found to change as a function of thalamic drive between 8 and 8.5Hz. […] As such, this non-linear phenomenon, which is well understood mathematically, does not impact our conclusions.”

6) Figure 3 shows a nice representation of the spectrum during the stimulation at 11 Hz over different levels of thalamic drive. It would be good to see the same figure without the stimulation in order to have an overview of the changes in the spectrum caused by the thalamic drive alone. Additionally, it would be good to see what the EEG spectrum looks like under these different levels of thalamic drive as in rest and task (in Figure 4, without stimulation).

We have now added both these excellent suggestions to the Figure 3 and Figure 4.

7) "Overall, our findings suggest that modulating brain oscillations is best achieved in states of low endogenous rhythmic activity," Shouldn't that say, "in states of low endogenous alpha activity"? Or do the authors think this holds for other frequencies as well? And if so, which and why, and where is the evidence for such a general claim?

In this context, we indeed refer to alpha activity. However, in a broader context (and as discussed in the Discussion section), our results support the idea that control over neuronal activity is better achieved in absence (or low levels of) intrinsic activity correlations (rhythmic attractor or otherwise). This perspective is supported by recent results in anesthesia (Greenberg et al., 2008) and seminal work on neural coding (Zohary et al., 1993) showing that increased correlations in neural systems hinder information transmission. As such our statement represents a perspective in the light of converging evidence, but remains – at this stage – only a suggestion. This is a topic of current interest for us and we seek to investigate it further. If the referees agree, we have relaxed these claims in the revised manuscript to reflect their rather conjectural nature.

We have first added in the Discussion:

“Formulated in the context of neural systems, irregular and/or uncorrelated fluctuations can thus be used to increase the saliency or fidelity of neural responses. […] Future work will show whether endogenous correlated neural activity in general, not only the rhythmic alpha activity observed here, could also play a role in regulating susceptibility of neural systems to entrainment, and more generally, control.”

And then relaxed our claim:

“In such work, our findings suggest that states of low alpha oscillations represent the ideal state for the application of stimulation and that modulation of brain oscillations is best achieved in states of low endogenous alpha rhythmic activity. This finding may be important for the next generation of brain stimulation paradigms developed using rational design.”

8) It would be helpful to the reader if the authors could clearly define what they mean by "state", e.g.:– Results section: "or other sensory inputs meant to fluctuate as a function of state"– Figure 1 legend: "Sensory inputs, whose intensity scales with state"Also later when discussing "state-dependent effects", use of the term "state" is ambiguous. Do the authors exclusively mean externally imposed experimental state (rest vs task condition)? Or do they also consider intrinsic (spontaneous?) brain state? Or both? Especially considering these two are not independent (e.g., spontaneous state might affect the response to external input), this should be discussed. In that light: "Visual inputs (or other sensory inputs meant to fluctuate as a function of state) are sent through the LGN to cortical areas for further processing". Do the authors mean to imply that thalamic input is unmodulated? i.e., does LGN, in this model, simply relay visual input? Or is LGN activity also modulated based on intrinsic brain state? All of this should be unambiguously spelled out in the manuscript. It would also be helpful to have the operational definition of task and rest states ("the rest state – which we have defined as a regime of weak inputs to the thalamus, and the task state – where thalamic drive is high") in the Introduction or early in the Results section.

Agreed. In our work, we have indeed considered that sensory drive to the thalamus (which mediates the suppression of endogenous alpha oscillatory activity) is not modulated perse but highly dynamic (noisy process with stationary statistics). By “state” we here refer to different sensory-drive conditions in which endogenous activity is significantly different in each case (see Figure 3). Our claim is also that this “state” is linked to the magnitude of thalamic drive. Sub-cortical population do not simply “relay” sensory activity but a reciprocally connected to cortical populations through a feedback loop and adjust their activity based on this sensory afference. Our model must be improved in the future so that attentional effect and other “self-driven” fluctuations in brain state are not solely linked to sensory/thalamic drive. This is a topic of active research by the authors.

To clarify what we mean by “brain state”, we have added the following in the Introduction:

“We defined regimes of low (resp. high) thalamic drive as rest (resp. task-engaged) brain state to represent two externally imposed experimental conditions.”

We have also added at the very beginning of the Results section:

“We note that by brain state, we here refer to the intensity of sensory drive and the resulting change in neural activity due to this drive, corresponding to externally imposed experimental state, not spontaneous fluctuations due to attention or other internal fluctuations.”

As well as the following comments:

“We did this to define quantitatively different brain states as a function of input to the thalamus”

“We thus defined rest and task states as limit cases of low (i.e. rest) and high (i.e. task) thalamic drive, respectively (see Materials and Methods), as indicated in Figure 2”.

To further clarify this point, we have changed the title of the section in the Materials and methods to “Thalamic Drive and Brain States” and added clear, bullet point operational definitions for the rest and task-engaged states.

9) In addition to a qualitative representation, a quantitative approach specifying the extent to which the proposed model is statistically robust could make it a much stronger contribution. The authors should substantiate their claims, e.g., focusing on the results presented in Figure 3: for which thalamic drive values is spike rate correlation significantly different for stimulation vs sham? Similarly, when is 8-Hz power sig different from 11-Hz power? Same for the phase-locking analysis. Figure 3 shows an asterisk for the peak of mutual information, is that signifying any statistical test result? All of this should be quantified using appropriate statistics, and documented in the Results section. (Same applies to results presented in the following figures.)

This is a very relevant point that we have improved in our revised manuscript. The asterisk was there to “identify” the peak position, not the statistical significance, leading to confusion. We have now quantified trial-to-trial variability and statistical significance (p-value) in our revision. Notably, we have redone the mutual information calculations completely (we add here that the MI calculations details in the Materials and methods section were accidentally cut in our initial submission. We apologize and this has been corrected). We now quantify the mutual information using spectral covariance and using an input signal of higher intensity (S=0.15, same as in all the panels). The following figures display simulation results for specific cases already quantified in Figure 3 (the rest and task states in Figure 5 and Figure 6), to illustrate the phenomenon in a single trial (Figure 4) or to explain the ideas underlying the theoretical framework (Figure 7).